# Molecular Pathogenesis of Avian Splenic Injury Under Thermal Challenge: Integrated Mitigation Strategies for Poultry Heat Stress

**DOI:** 10.3390/cimb47060410

**Published:** 2025-05-31

**Authors:** Qing Liu, Lizhen Ma, Lili Liu, Ding Guan, Zhen Zhu, Xiangjun Hu

**Affiliations:** 1School of Life Science and Food Engineering, Hebei University of Engineering, Handan 056038, China; qingliu670214@163.com (Q.L.); liull0728@163.com (L.L.); guanding2343@163.com (D.G.); 2Beijing Institute of Radiation Medicine, 27 Taiping Road, Beijing 100850, China; malizhen_2022@163.com

**Keywords:** heat stress, chickens, spleen, immunological damage, intervention strategies

## Abstract

Heat stress (HS), an important environmental stressor for healthy poultry farming, has been shown to have a detrimental effect on production performance and induce serious diseases through immune system damage. As the avian peripheral immune system’s primary organ, spleen is subject to complex biological processes in response to HS injury. Histopathological characterization demonstrated that HS resulted in the destruction of the splenic red and white medulla, a decrease in cell density and organ atrophy. These changes directly impaired pathogen clearance and immune surveillance. At the physiological level, the impact of HS is characterized by disrupted metabolic homeostasis through interrupting neuroendocrine function. This, in turn, results in a significant suppression of humoral immune response. The oxidative-inflammatory cascade constitutes the core pathology of this disease. Energy metabolism disorder triggered by mitochondrial dysfunction and redox imbalance form a vicious circle, which promotes apoptosis signaling cascade. Meanwhile, over-activation of intrinsic immune system triggers a series of inflammatory factors, which further amplifies effects of tissue damage. The present prevention and control strategies are centered on synergistic anti-inflammatory and antioxidant interventions with nutrient modulators and plant actives. Nevertheless, it is imperative for future studies to incorporate multi-omics technologies in order to analyze the metabolic mechanisms and patterns of stress and establish a precise intervention strategy based on immune homeostatic regulation. This review systematically investigated the multilevel regulatory mechanisms of HS-induced spleen injury, which provides a theoretical basis for the mechanistic analysis and technological innovation of the prevention and control of HS syndrome in poultry.

## 1. Introduction

Heat stress (HS) is a major environmental challenge for the poultry industry, especially in the context of climate change. The effects of HS are becoming increasingly significant, and research on the hazards of HS and its prevention and control is, therefore, becoming increasingly important. In the context of elevated ambient temperatures, the poultry industry faces a multifaceted challenge, with detrimental effects on production performance. These include a decline in feed intake, delayed weight gain [1], reduced egg production, and deteriorating eggshell quality [2]. Additionally, reproductive performance in breeders is impaired, leading to a simultaneous decline in both fertilization and hatchability rates [3]. Concurrently, the chickens’ immune systems are compromised, rendering them more vulnerable to Newcastle disease, E. coli and other diseases [4]. This results in the onset of respiratory alkalosis [5], oxidative damage and other metabolic imbalances [6]. In severe cases, these imbalances can directly precipitate acute heat exhaustion, leading to elevated mortality rates in large-scale poultry farms in the short term [7]. The harm caused by HS to the chicken industry is multifaceted, resulting in significant economic losses for farmers.

It is evident that HS can exert an influence on the growth performance of poultry through multiple pathways. Studies demonstrated that HS can lead to a reduction in feed intake (FI) and body weight gain (BWG) in broilers and laying hens, concurrently resulting in a decrease in feed conversion ratio (FCR) [8,9]. This phenomenon may be attributed to HS-induced imbalances in neuroendocrine regulation, such as the activation of the hypothalamic–pituitary–adrenal (HPA) axis, which triggers an increase in corticosterone levels, thereby suppressing growth hormone axis function [10,11]. Furthermore, HS has been demonstrated to disrupt the morphological structure of the gut, leading to a reduction in villus height and an increase in crypt depth, significantly impairing the ability to absorb nutrient uptake [12]. Furthermore, the repercussions of HS extend to the reproductive system, manifesting as a decline in blood ionized calcium (iCa) levels in laying hens and a decrease in Hutchinson’s units, consequently leading to impaired eggshell and protein quality [8]. Additionally, it has been observed that HS can impair germ cell mitochondrial function, thereby triggering reduced fertilization rates and abnormal embryo development through the process of oxidative stress [13]. Prolonged exposure to HS has also been demonstrated to alter epigenetic regulatory mechanisms, which may consequently affect the production potential of offspring [11].

The spleen is a vital organ in the avian immune system, and its function is significantly impacted by HS conditions. Research has demonstrated that HS disrupts the antioxidant defense system, significantly reducing superoxide dismutase (SOD) and Glutathione Peroxidase (GPx) activities, leading to malondialdehyde (MDA) accumulation, inducing lymphocyte apoptosis [14,15] and inhibiting immune organ development [15]. Furthermore, corticosterone levels, elevated through the HPA axis, have been demonstrated to reduce antibody production and disrupt T/B lymphocyte homeostasis [10,16]. Chronic heat stress (CHS) has been demonstrated to trigger inflammatory responses in the spleen and to upregulate pro-inflammatory cytokine expression through the activation of TLRs/MyD88/NF-κB signaling pathways [17]. This, in turn, exacerbates splenic injury, leading to structural damage of the red and white pulp in the spleen and impaired immune function [17]. Furthermore, the proliferation of intestinal pathogenic bacteria is promoted by HS, which exacerbates systemic immunosuppression through the gut–spleen axis [14], and consequently significantly increases the susceptibility of poultry to pathogens [18], thereby posing a threat to the biosecurity of the farming industry. Consequently, there is a necessity for in-depth research to be conducted on the mechanisms of HS damage to the spleen, in order to develop effective anti-stress strategies that will improve poultry performance and disease resistance.

In recent years, studies have demonstrated that HS can cause pathological damage to the spleen through a number of mechanisms, including oxidative damage, apoptosis and inflammation. This, in turn, can lead to immunosuppression and reduced growth performance. However, the specific molecular mechanisms through which these effects occur have not yet been fully elucidated. This paper systematically describes the dynamic effects of HS on the physiological function, tissue structure and molecular signaling pathways of the chicken spleen. It integrates the progress of nutritional regulation and other interventions, with the aim of providing a theoretical basis for the analysis of the cascade mechanism of HS-induced splenic injury. The paper also provides a scientific reference for the development of targeted mitigation strategies (e.g., the addition of antioxidants and the application of immune-enhancing agents).

## 2. Effects of HS on the Organization of the Chicken Spleen

HS in poultry can be generally classified into two categories: acute and chronic. Acute heat stress (AHS) refers to short-term exposure (from minutes to days, typically ≤72 h) to extreme ambient temperatures exceeding the thermoneutral zone, which primarily induces cardiovascular failure and acute mortality in chickens. In contrast, CHS involves prolonged exposure (from weeks to multiple generations) that leads to systemic metabolic disturbances, particularly in energy homeostasis and oxidative balance, ultimately resulting in sustained depression of productive performance [6,19].

Histopathological studies have demonstrated that HS (acute and chronic) can induce multidimensional structural damage in the chicken spleen [20,21]. Hirakawa et al. [22] observed that HS for 14 days at 34.5 °C resulted in hypoplasia of the splenic germinal center and morphological changes in the thyme cortex and bursal follicles. These observations confirm that HS disrupts the avian microenvironment for the development of the central immune organs. Chen et al. [17] further demonstrated that AHS (35 °C, 10 h/d) in 42-day-old broilers resulted in a blurring of the red-white medullary boundary, a decrease in the number of parenchymal cells, and a reduction in the size of the spleen. Concurrently, a significant decrease in the spleen index (spleen weight/body weight ratio) was observed, accompanied by persistent activation of the TLRs/MyD88/NF-κB signaling pathway. These findings suggest that impaired integrity of the immune microenvironment is associated with over-activation of the inflammatory response. Hui et al. [23] observed a reduction in splenic white medulla oblongata and disorganization of lymphoid node structure in a murine model, which exhibited similar pathological features to the blurring of red and white medulla oblongata boundaries in avian species.

## 3. Physiological Effects of HS on Chicken Spleen

### 3.1. Thermoregulatory and Metabolic Changes

Chickens are characterized by a narrow core body temperature range (41–42 °C) and an absence of sweat glands, which considerably restricts their thermoregulatory capacity during periods of HS. At the physiological level, chickens primarily dissipate heat through respiratory evaporation and peripheral vasodilation. Kim et al. [24] found that respiratory water evaporation from the respiratory tract of chickens significantly increased under AHS by respiratory rate monitoring, and the respiratory rate sharply increased from 25 to 180 breaths/min, which resulted in rapid cooling. Al-Ramamneh et al. [25] and Brugalletta et al. [26] found that HS-induced vasodilatation significantly increased blood flow to chicken skin and mucous membranes (especially to the crown and beard area), but triggered insufficient blood supply to internal organs such as the intestine and tissue hypoxia. Li et al. [27] demonstrated that HS activated the hypothalamic–pituitary–adrenal axis in chickens, which prompted an increase in the secretion of adrenocorticotropic hormone and corticosterone, leading to a preferential allocation of body energy to heat dissipation processes instead of to the growth or the immune system. Liu et al. [28] found that HS-induced calcium disruption accelerated the process of apoptosis, thereby negatively affecting meat quality. Fathi et al. [29] revealed through genotype–phenotype association analysis that chicken lines carrying either the naked neck gene or the dwarf gene showed superior adaptations to commercial broilers in terms of heat tolerance by enhancing the efficiency of heat dissipation through the skin. Sandercock et al. documented through behavioral observations that chickens briefly regulate during HS through behaviors such as reduced activity and wing spreading, but were unable to circumvent the onset of cumulative organ damage.

It is evident that HS significantly interferes with organismal metabolic homeostasis. Gonzalez-Rivas et al. [30] found blood glucose levels in chicken livers during the initial phase of AHS revealed a transient elevation of blood glucose, which can be attributed to hepatic glycogenolysis. This team also observed the inhibition of mitochondrial function and reduced metabolic efficiency in a model of CHS. Lu et al. [31] examined and analyzed chicken pectoral muscle tissues and found that enhanced lipolysis was accompanied by malondialdehyde accumulation and mitochondrial dysfunction. The enhanced lipolysis was accompanied by malondialdehyde accumulation and mitochondrial dysfunction. Ma et al. [32] demonstrated that HS contributed to amino acid redistribution to the hepatic gluconeogenesis pathway by up-regulating muscle atrophy-related genes, such as Muscle Atrophy F-box protein (*MAFbx*), in skeletal muscle. Habashy et al. [33] and Ma et al. [32] found that corticosterone enhanced gluconeogenesis through the enhancement of key enzymes, such as Phosphoenolpyruvate Carboxykinase (PCK), Fructose-1,6-bisphosphatase1(FBP1) and other gluconeogenic enzymes, in the chicken liver. Gluconeogenesis key enzyme activities, while inhibiting the expression of nutrient transport proteins such as intestinal Peptide Transporter 1 (PepT1) and Fatty Acid Transport Protein 1 (FATP1), leading to limited energy substrate supply. Seo et al. [34] utilized microbiome sequencing and host gene expression profiling to ascertain that HS altered the jejunum flora of broiler chickens (with a decrease in the genus Anabaena) and up-regulated fatty acid oxidation genes, such as *Acyl-CoA dehydrogenase, long chain* (ACADL) and *Acyl-CoA oxidase* (ACOX), thereby activating host-microbe interactions. Hirakawa et al. [22] detected heat-stressed broiler spleen immune cells by immunofluorescence staining and found a significant reduction in the number of Bu1+ B cells, CD3+ T cells (including CD4+ and CD8+ subsets), and a significant decrease in the number of biomass. The study also observed significant reductions in the number of CD3+ B cells and developmental defects in the germinal center. The authors hypothesize that these phenomena may be attributable to an imbalance in physiological homeostasis triggered by inadequate supply of glucose and fatty acids as a result of metabolic reprogramming, and by increased energy consumption in wheezing behavior.

### 3.2. Impaired Immune Function

Significant structural damage to the immune organs of broiler chickens (especially the spleen and bursa) has been observed under HS conditions [35]. The study found that spleen and bursa indices of chronically heat-stressed broilers showed reduced splenic germinal center formation and inhibition of the TLR4-TBK1 signaling pathway, leading to reduced interferon production, despite the elevated organ weight/body weight ratio [35]. Calefi et al. [36] observed a significant decrease in the number of lymphocytes in the spleen and duodenal lamina propriety of heat-stressed broilers via lymphocyte counts. Habibian et al. [37] and Sun et al. [38] observed a significant decrease in the number of lymphocytes in the spleen and duodenal lamina propria of heat-stressed broilers via lymphocyte counting. Habibian et al. [37] and Sun et al. [38] found a significant decrease in the relative weight of the thymus, with morphological alterations of the spleen and the bursa, and a delay in the development of the bursa associated with impaired immunoglobulin secretion in heat-stressed broilers. Sun et al. [38] further demonstrated that HS resulted in abnormalities in spleen tissue structure, and Bartlett et al. [39] and Habibian et al. [37] demonstrated in a comparative study that the above structural alterations collectively resulted in immunosuppression, as evidenced by decreased lymphoid organ weights and a diminished cellular immune response.

Lymphocyte subpopulation ratios and immune function also showed significant alterations in poultry under HS. Honda et al. [40] found a significant decrease in the proportion of B lymphocytes by examining the ratio of T/B lymphocyte subpopulations in the peripheral blood of chickens under HS. Rocchi et al. [41] observed a simultaneous increase in the proportions of cytotoxic T cells (CD8+) and helper T cells (CD4+) in the same model. Ju et al. [42] found a significant increase in peripheral blood CD8+ T cell ratio and a decrease in CD4+/CD8+ ratio on days 1 and 14 of HS by dynamic monitoring in a Parmesan miniature pig model, whereas Huo et al. [43] found instead a significant increase in the CD4+/CD8+ T cell ratio when examining the splenic T cell subset after CHS in pigs. The present study hypothesizes that immunosuppression may be associated with this ratio imbalance. At the level of molecular mechanisms, Huo et al. [43] found that HS upregulated the expression of the anti-apoptotic factor Bcl-2 in splenic lymphocytes, whereas Tang et al. [44] demonstrated that prolonged heat exposure reduced chicken splenic CD4+ T cell survival. In terms of immune function, Honda et al. [40] found that HS significantly decreased IgG antibody titers against Newcastle disease virus in vaccinated chickens. Niu et al. [45] found that HS inhibited phagocytosis and natural killer cell activity of chickens’ macrophages, and Rocchi et al. [41] further found that HS led to the impairment of innate immune defenses in chickens by pathogen attack experiments.

In a chicken model of HS, humoral immune indices demonstrated dynamic changes. Akinyemi and Adewole [46] observed that Ross 308 broiler chickens subjected to CHS (32–34 °C, 8 h/d, 7 d) exhibited significantly elevated serum IgG and IgM levels, but decreased complement C3 and C4 concentrations, which may be indicative of suppression of hepatic function. Hu et al. [47] further observed a decrease in complement C3 levels in heat-stressed broilers, along with suppression of innate immune-related enzyme activities, suggesting a common impairment of the complement system and the innate immune barrier. In terms of the regulatory mechanisms of cytokines, He et al. [48] reported that the mRNA expression of pro-inflammatory factors (e.g., IL-1β, IL-6, and TNF-α) was significantly elevated in the spleen of yellow-feathered broiler chickens, while IL-10 remained relatively unchanged. This observation suggests a potential imbalance between pro-inflammatory and anti-inflammatory responses. In a Salmonella infection model, Quinteiro-Filho et al. [49] observed that HS led to diminished vaccine-specific antibody responses and a reduction in mucosal immune-related antibodies (e.g., IgA). It is noteworthy that Hassan et al. [50] conducted a comparative analysis of various strains of laying hens, revealing substantial strain-specific variations in the response to AHS, particularly with regard to IgG concentrations. This observation suggests the potential for genetic background to influence the magnitude of humoral immune responses. Intervention studies have demonstrated that Li et al. [51] found that tryptophan supplementation improved immunoglobulin levels in heat-stressed broilers, whereas He et al. [48] proposed that resveratrol alleviated inflammatory factor hyper activation by inhibiting the NF-κB pathway. These findings collectively indicate that HS interferes with humoral immune homeostasis through multiple pathways, suggesting the need for targeted nutritional modulation strategies.

A comprehensive analysis of the physiological effects of HS on avian splenic tissue demonstrates that both AHS and CHS disrupt thermoregulatory mechanisms and metabolic homeostasis, whereas immune dysfunction manifests predominantly under CHS conditions.

## 4. Molecular Mechanisms of HS-Induced Splenic Injury in Chickens

### 4.1. Oxidative Stress

Oxidative stress is a pathological state resulting from an imbalance between the production of reactive oxygen species (ROS) and the antioxidant system. Yang et al. [52] demonstrated using a liver mitochondrial function assay that acute heat exposure (35 °C for 3 h) induced an increase in ROS production accompanied by a compensatory increase in SOD and GSH-Px activities, but antioxidant enzyme activities gradually returned to baseline levels within 12 h of stress relief. Similarly, Aengwanich and Wandee [53] found that in a model of AHS (34–36 °C), chicken hemocyte SOD activity showed a transient increase that was synchronous with the up-regulation of HSP70 expression, suggesting an early compensatory activation characteristic of the antioxidant defense system. When the stress became chronic, Chen et al. [17] observed a sustained decrease in total antioxidant capacity (T-AOC) and SOD activities and an increase in MDA levels in spleen tissue, as well as activation of inflammatory responses via the TLRs/MyD88/NF-κB pathway. The mechanism of this acute and chronic difference was systematically elucidated by Akbarian et al. [54]: acute stress is dominated by overload of the mitochondrial electron transport chain, whereas chronic stress is associated with downregulation of antioxidant gene expression and atrophy of mitochondrial metabolic capacity. Membrane system damage is directly linked to lipid peroxidation. Feng et al. [55] found in an in vitro muscle mitochondrial model that high-temperature incubation at 44.5 °C not only significantly increased H_2_O_2_ and MDA content, but also inhibited Ca^2+^-ATPase activity, leading to a loss of mitochondrial membrane potential. This mechanism of injury may be further amplified in the spleen, where Brannan et al. [56] showed that AHS (32 °C, 4 h/d, 1 d) significantly impaired thyme mitochondrial ROS scavenging by down-regulating avian uncoupling protein (avUCP) expression, with male chickens being more sensitive to this pathway.

Oxidative stress regulation by small heat shock proteins (sHSPs) exhibits significant tissue-specific features. In a black-boned chicken model, Liu’s team [57] found that HSP27 and HSP90 mRNA expression was significantly suppressed in thymus tissue during the early phase of AHS (37 °C, 15 days), whereas splenic HSP70 was specifically overexpressed, revealing a pattern of differential regulation between organs. This spatiotemporal specific regulation directly affects cell fate decisions: Aengwanich and Wandee [53] demonstrated that 42–43 °C treatment retarded endogenous apoptotic processes by inhibiting caspase-9/3 pathway activity through HSP70-mediated mitochondrial membrane stabilization. However, overcompensation may breach the homeostatic threshold. Mao et al. [58] revealed that splenic HSP70/90 overexpression triggered by short-term hyperthermia (38 °C, 4 h) abnormally activated the release of proinflammatory factors, such as IL-1β and IFN-γ, creating an amplified effect of oxidative stress-inflammation cascade through an intervention experiment with Zygomycetin. Notably, there is a multi-system synergy in compensatory regulation. Wang’s team [59] found that subacute heat stress (SAHS) (36 °C, 8 h/day) enhances the transmembrane transport efficacy of antioxidants through the up-regulation of ATP-Binding Cassette Subfamily G Member 2 (ABCG2) and Sodium-Dependent Vitamin C Transporter 2 (SVCT-2) expression in immune organs, suggesting that a compensatory defense network exists in the organism.

### 4.2. Apoptotic Pathways

HS induces apoptosis in chicken splenic lymphocytes through multi-pathway interactions, and its core mechanism revolves around the mitochondria-dependent apoptosis pathway. Zhang et al. [60] and Xu et al. [61] showed that HS significantly increased ROS levels in splenic tissues, leading to a decrease in the mitochondrial membrane potential and impaired activity of Complex I/III, which then triggered a decrease in ATP synthesis and accumulation of oxidative stress markers such as MDA. This synergistic effect of mitochondrial dysfunction and oxidative stress constitutes the starting signal of the apoptotic cascade.

HS initiates programmed cell death via the mitochondria-dependent apoptotic pathway. Studies have confirmed [61,62] that the process begins with a perturbation of the dynamic balance of Bcl-2 family proteins—HS significantly upregulates the expression of the pro-apoptotic protein Bax while suppressing the production of the anti-apoptotic protein Bcl-2, resulting in a fold increase in the Bax/Bcl-2 ratio. This key molecular switch [61,63] directly induces an abnormal opening of the mitochondrial permeability transition pore (mPTP), which triggers the efflux of cytochrome C (Cyt c) into the cytoplasm where it binds to Apoptotic Protease-Activating Factor 1 (APAF-1) to form an apoptosome complex, which then cascades into the activation of caspase-9 and caspase-3, ultimately triggering DNA breakage and other irreversible apoptotic effects such as DNA degradation. At the upstream regulatory level, the p53 signaling pathway is a central regulatory hub [61,64]. HS activates p53 to achieve a dual pro-apoptotic mechanism: (1) transcriptionally promoting Bax gene expression and enhancing pro-apoptotic signals; (2) inhibiting Bcl-2 biosynthesis and weakening the mitochondrial protective barrier, resulting in a bi-directional amplification of the regulatory effect.

In addition to the mitochondrial pathway, HS injury to the spleen is closely associated with NF-κB-mediated inflammatory responses. Ma et al. [65] showed that HS significantly upregulated the phosphorylation level of NF-κB p65 in the spleen and promoted the secretion of pro-inflammatory factors IL-6 and TNF-α. Furthermore, animal responses to HS are species-specific: Xu et al. [61] observed more severe mitochondrial ultrastructural damage (e.g., cristae breaks) in chicken spleens than in mammals, and a higher degree of suppression of innate immune pathways such as TLR/NOD. This difference has been further confirmed in cross-species studies: Liu et al. [66] found that the Nrf2/Keap1 pathway dominated the activation of the antioxidant system in a pig model, whereas Wang et al. [67] found that apoptosis of mouse splenic lymphocytes was mainly dependent on the p38 MAPK pathway rather than the mitochondrial pathway, suggesting an evolutionary divergence in the mechanisms of response to HS in different species.

Several studies have proposed intervention strategies based on the apoptotic pathway to address the above mechanisms. Regarding the regulation of apoptosis, Meng et al. [62] found that resveratrol significantly attenuated apoptosis in chicken splenic lymphocytes by upregulating Bcl-2 expression and inhibiting caspase-3 activity. On the other hand, regulation of the inflammatory response can be achieved by inhibition of NF-κB p65 phosphorylation, as Liu et al. [68] found that seaweed polysaccharides could reduce the release of pro-inflammatory factors and, thus, improve spleen immune function. In addition, Liu et al. [66] showed that hydroxyl selenomethionine (SeO) could enhance antioxidant enzyme activities and alleviate HS-induced oxidative damage by activating the Nrf2/Keap1 pathway. Interestingly, Badr et al. [69] found that camel whey protein (CWP) could downregulate HSP70 expression and restore T/B cell distribution. The diversity of these interventions not only reflects the multi-target regulatory potential of the apoptotic pathway, but also provides a theoretical basis for optimizing HS management strategies in avian.

### 4.3. Inflammatory Signaling Pathways

Chen’s team [17] found that HS significantly activated inflammatory responses in the chicken spleen via the TLR2/4-MyD88-NF-κB signaling axis, which manifested as an imbalance between elevated Th1-type cytokines (IL-6, TNF-α) and suppressed Th2-type factors (IL-4). This classical pathway has a unique complementary mechanism in avian species: Ma et al. [70] found that chicken spleen specifically up-regulated DNA-dependent activator of interferon (DAI) gene expression, suggesting the presence of non-classical pattern recognition receptor-mediated inflammatory signaling. Ding et al. [71] further demonstrated in a combined stress model (rostral pruning and HS) that high MyD88 gene expression was significantly positively correlated with inflammatory markers such as IL-1β and NF-κB, and TLR4 signaling had a synergistic amplification effect.

In particular, Nanto-Hara et al. [72] showed that HS exacerbates the inflammatory microenvironment through a dual mechanism: on the one hand, it significantly reduces the activity of antioxidant enzymes (SOD, CAT), and on the other hand, it promotes the accumulation of the marker of oxidative damage, MDA, resulting in a vicious oxidative-inflammatory cycle. Comparative studies have shown that the TLR4-MyD88-NF-κB pathway is highly conserved in mammals: Feng et al. [73] found in a porcine diabetes model that TLR4 activates IRAK-1/NF-κB via a MyD88-dependent pathway and also enhances the interferon response via the TRIF-IRF3 branch. Mouse experiments [74] further confirmed that NF-κB activation triggered neuroinflammation in the hippocampal region, accompanied by elevated IL-6, TNF-α and cognitive impairment. These findings suggest that the TLRs-NF-κB axis is a core pathway for HS inflammation across species, but that avian species may complement the signaling network with unique receptors such as DAI.

HS-induced HSP70 exhibits significant bidirectional regulatory features in avian and mammalian species. Chen et al. [17] found that chicken spleen HSP70 has spatiotemporal specificity: it exerts anti-inflammatory protection in the acute phase by inhibiting IκBα degradation to limit NF-κB overactivation, but enhances MyD88 signaling through TLR4 endocytosis in response to chronic stress, resulting in persistent inflammation. This paradoxical effect has been demonstrated in mammals: Lee et al. [74] found that in a mouse model, HSP70 persistently activates the TLR4-NF-κB axis, triggering glial cell activation and neuronal apoptosis in the hippocampus. Gouda’s team [75] suggested that the functional switch of HSP70 may depend on the kinetics of its expression, with a protective role in acute stress and a shift to pro-inflammatory effects with chronic stress. In response to the oxidative-inflammatory cascade induced by HS, researchers have developed multi-targeted intervention strategies: Liu et al. [66] found that hydroxy-selenomethionine effectively alleviated oxidative damage in porcine spleen by activating the Nrf2/Keap1 pathway while inhibiting NF-κB and STAT3 signaling. Ma et al. [70] demonstrated that peppermint extract reduced IF-κB and HSP70 by specifically inhibiting the avian DAI-NF -κB axis and reduced levels of inflammatory factors such as IL-6 and TNF-α. Although existing interventions have focused on the oxidative-inflammatory cross-reaction, the avian-specific DAI regulatory mechanisms remain to be thoroughly analyzed, which may be a key breakthrough in the development of species-specific control strategies.

There are significant differences between avian and mammalian responses to HS. Nanto-Hara et al. [72] found that the oxidative stress response in chicken spleen was more sensitive to HS and that HS resulted in greater decreases in SOD and CAT activities than in mammals. In addition, the specific role of the DAI gene in avian inflammation may replace some of the TLR4 function [70], while activation of the TLR4-TRIF-IRF3 pathway in a pig model suggests the importance of the non-classical NF-κB branch in mammals [73].

## 5. Strategies for Mitigating HS Splenic Injury

HS causes structural and functional damage to the spleen through oxidative stress, apoptosis and activation of inflammatory signaling pathways [70]. In response to these studies, natural products and nutritional supplements have shown potential for intervention, as shown in Table 1. Peppermint extract (Mentha piperita) significantly reduced splenic pro-inflammatory factor levels and ameliorated pathological tissue damage in the spleen by inhibiting the DAI (DNA-dependent interferon regulatory factor activator)-mediated NF-κB and IRF3 signaling pathways [70]. Alzarah et al. found that *Citrullus colocynthis* seeds could alleviate HS-induced oxidative damage and inflammatory response in the spleen by increasing T-AOC and downregulating TNF-α expression [76]. Guanidinoacetic acid (GAA) indirectly improves splenic lymphocyte activity and increases the thymus and bursa immune organ index by modulating the HPA axis and decreasing corticosterone (CORT) levels [27]. On the other hand, gamma-aminobutyric acid (GABA) and postnatal bacteria (such as *Lactobacillus plantarum* RI11) indirectly support spleen function by modulating the gut-immune axis and enhancing the systemic immune response [77,78].

The synergistic application of thermal adaptation training and genetic improvement holds promise for preventing HS-induced splenic injury in poultry. Studies demonstrate that neonatal chicks exposed to repeated thermal conditioning (40 °C, 15 min/d) during the critical 2–4 days postnatal period exhibit suppressed stress hormone secretion and modulated hypothalamic gene expression, thereby enhancing physiological adaptability to acute hyperthermia. Notably, cumulative effects induced by multiple thermal stimuli surpass the efficacy of single interventions [79]. This programmed heat acclimatization aligns with the evolutionary principles of Bergmann’s and Allen’s rules: climatic selection pressures drive organisms to enhance thermal tolerance through morphological adaptations (e.g., optimized surface heat dissipation) and molecular mechanisms (e.g., metabolic reprogramming) [80].

Genomic evidence further identifies positive selection signatures in tropical chicken breeds for SLC33A1 and TSHR genes. The former regulates endoplasmic reticulum acetylation to maintain cellular homeostasis, while the latter optimizes glucose metabolism and mitochondrial function via thyroid hormone signaling pathways. Their low-frequency mutant alleles may serve as molecular markers for heat-resistant breeding strategies [81]. Cross-species studies on the HSP70 pathway in Brahman cattle reveal that multigenerational selection of heat-tolerant genotypes enhances systemic thermal resilience, though genotype-environment interactions affecting production traits require careful evaluation [82]. Ravagnolo and colleagues (2000) established an inverse correlation between Holstein milk yield and temperature-humidity index [83].

Integrating early-life thermal adaptation protocols with targeted gene screening (e.g., heat shock proteins and metabolic regulators [84]) may enable the development of climate-resilient poultry populations while minimizing productivity losses.

## 6. Outlook

The physiological effects of HS on the chicken spleen are mainly reflected in metabolic changes and impaired immune function. HS activates the hypothalamic–pituitary–adrenal axis in chickens, leading to an increase in hormone secretion, which results in a preferential allocation of energy to heat dissipation processes rather than to growth or the immune system. HS also disrupts metabolic homeostasis, affecting blood glucose levels, mitochondrial function, and lipolysis and gene expression, resulting in a limited supply of energy substrates. In terms of impaired immune function, significant structural damage occurs in broiler immune organs under HS conditions, such as changes in spleen and bursa indices, reduced germinal center formation and suppression of the TLR4-TBK1 pathway. Meanwhile, humoral immune indices of heat-stressed poultry also showed significant dynamic changes, such as increased serum IgG and IgM levels and decreased complement C3 and C4 concentrations. The negative effects of HS on chicken spleen and immune function have been widely studied, but the molecular mechanisms, genetic regulatory networks and intervention strategies still need to be explored in depth. Future studies should focus on multi-target synergistic intervention strategies. Metabolomics-driven screening of key regulators (e.g., DAI, Nrf2, TLR4) may provide a theoretical basis for the development of novel functional additives [70,85]. In addition, the combined use of natural products and dietary supplements (e.g., peppermint extract and guanidinoacetic acid) may be more effective in alleviating the combined injury of HS on the spleen through the synergistic effect of antioxidant, anti-inflammatory and immunomodulatory effects [27,76]. Based on the existing research results, future studies can be carried out in the following directions:Investigating the mechanism of HS: Using single-cell sequencing and other technologies, we will investigate the effect of HS on the subpopulation of immune cells in chicken spleen, and clarify the specific changes of different immune cells and their interaction mechanisms under HS. At the same time, the molecular mechanism of HS on chicken spleen immune function, in particular the relationship between immunosuppression and the balance of the cytokine network, should be researched in-depth to provide more precise targets for subsequent interventions.Research on strategies to mitigate the effects of HS: On the one hand, in-depth research on nutritional regulation strategies, screening and validation of more nutrients that can mitigate the effects of HS on chicken immune function, such as functional amino acids, vitamins, minerals, etc., and explore the mechanism of their action. On the other hand, in combination with environmental control measures, such as optimizing the ventilation and cooling system of the chicken house, we will comprehensively evaluate the effects of different measures to mitigate the effects of HS, to provide a basis for the development of scientific and reasonable feeding and management plans in actual production.Research on heat resistance of different chicken breeds or genotypes: Further expand the scope of research, screen and identify more chicken breeds or genotypes with excellent heat resistance, and thoroughly analyses the genetic basis and molecular mechanism of their heat resistance, so as to provide theoretical support for chicken breeding and to breed better breeds that are better adapted to high temperature environments. However, traditional genetic improvement approaches may compromise productivity, necessitating the integration of epigenetic mechanisms and molecular biology techniques to achieve equilibrium between thermotolerance and production metrics.

## Figures and Tables

**Table 1 cimb-47-00410-t001:** Summary of the changes in indexes, key pathways, and mitigation strategies of the organization structure, physiological effects, and molecular mechanisms of the spleen in chickens under heat stress (HS).

	Indicator Changes	Key Pathways	Mitigation Strategies
Organization Structure	Spleen index ↓ [17] Number of parenchymal cells ↓ [17]	--	--
Physiological Effects	Respiratory rate ↑ [24] Vasodilation [25,26]	Hypothalamic–Pituitary–Adrenal Axis [27]	GAA [27]
	Adrenocortical Hormones ↑ [27] Cortisol ↑ [27] Blood glucose ↑ [30]		
	Splenic germinal center formation ↓ [35] Splenic lymphocytes ↓ [36]	TLR4-TBK1 [35]	Tryptophan [51] GABA [78]
	Proportion of B lymphocytes ↓ [40]		
	IgG ↑ [46] IgM ↑ [46] C3 ↓ [47]		
Molecular Mechanism	ROS ↑ [52] SOD ↑ [53]/↓ [17]	Mitochondrial electron transport chain [54]	SeO [66] CWP [69] Citrullus colocynthis seeds [76] Resveratrol [48] Seaweed Polysaccharides [68] Mentha piperita [70]
	Bax/Bcl-2 ↑ [61,62]	Mitochondria-dependent apoptosis pathway [60,61]	
	IL-4 ↓ [17], IL-6 ↑ [17], TNF-α ↑ [17], DAI ↑ [70]	TLR2/4-MyD88-NF-κB [17]	

^1^ ↑: Increase, ↓: Decrease.

## Data Availability

No new data were created or analyzed in this study. Data sharing is not applicable to this article.

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
