# Peer review of "Molecular Pathogenesis of Avian Splenic Injury Under Thermal Challenge: Integrated Mitigation Strategies for Poultry Heat Stress"

_cimb, 2025, doi:10.3390/cimb47060410_

Round 1
Reviewer 1 Report
Comments and Suggestions for Authors
Below are detailed suggestions:
- It suggest that the acute heat stress and chronic heat stress can be discussed separately in each section based on their unique physiological impacts, molecular mechanisms, and downstream effects, and even according to the different species of poultry. This structured presentation will reinforce the article’s logical framework and facilitate better comprehension.
- In the context, the overall grammar is standardized and the academic expression is rigorous, while there are also some minor errors exist. Such as line 44, 80-81, 156, and 198-199.
- In the part 2, it suggest that acute heat stress and chronic heat stress can be simply defined first, and then the histological changes of the chicken spleen under different heat stress conditions can be introduced respectively.
- Table 1, the current table only lists the indicator changes and intervention strategies. It is suggested to add a column to briefly explain the key molecular pathways.
- Line 250, the acute heat stress (32℃ for 4h) seems to in line with the definition of the chronic heat stress (32-34℃, 8h/d), in the line 205. So how did you think of this issue?
In the context, the overall grammar is standardized and the academic expression is rigorous, while there are also some minor errors exist. Such as line 44, 80-81, 156, and 198-199.
Author Response
Reviewer(s)' Comments to Author:
Reviewer: 1
1) It suggest that the acute heat stress and chronic heat stress can be discussed separately in each section based on their unique physiological impacts, molecular mechanisms, and downstream effects, and even according to the different species of poultry. This structured presentation will reinforce the article’s logical framework and facilitate better comprehension.
A1): Thank you for your suggestions. We consider that not every part is suitable to be divided into acute and chronic. At the end of the third part, we add a general difference about the physiological effects of acute and chronic heat stress on chicken spleen.
2) In the context, the overall grammar is standardized and the academic expression is rigorous, while there are also some minor errors exist. Such as line 44, 80-81, 156, and 198-199.
A2: Thanks for the suggestion. These have been revised in the revised manuscript. See line 43-45, 78-80, 164-165, and 206-207.
3) In the part 2, it suggest that acute heat stress and chronic heat stress can be simply defined first, and then the histological changes of the chicken spleen under different heat stress conditions can be introduced respectively.
A3: Thanks for the suggestion. We have added relevant content in the revised manuscript. See Line 99-106.
4) Table 1, the current table only lists the indicator changes and intervention strategies. It is suggested to add a column to briefly explain the key molecular pathways.
A4: Thanks for the suggestion. We have added relevant content in the revised manuscript. See Table 1.
5) Line 250, the acute heat stress (32℃ for 4h) seems to in line with the definition of the chronic heat stress (32-34℃, 8h/d), in the line 205. So how did you think of this issue?
A5: Thanks for the suggestion. I didn't express my meaning clearly before. The correct content can be viewed in the manuscript. See line 213,262.
Reviewer 2 Report
Comments and Suggestions for Authors
- This article is based on the current problems caused by increasingly long and intense periods of high air temperatures in many areas of the world. From this perspective, poultry breeding and fattening are becoming increasingly demanding in terms of deeper understanding of the related issues and the use of new knowledge in practice.
- The article is written in a clear manner and brings a number of new findings that may be very useful for readers of the Animals journal who deal with the given issue or are focused on the related sectors of poultry breeding and animal production.
- I particularly appreciate part 5. Strategies for mitigating heat stress splenic injury. I would recommend that the authors try to supplement and further develop this important part of their article in the future.
- Similarly, in my opinion, the last part of the article 6. Outlook is also very valuable, in which the authors indicate the direction in which further development and research in this area should take place.
- Overall, I evaluate this article very positively as beneficial for the given scientific field and related fields. Heat stress in a changing environment also affects wild birds, which can survive even in a very hot climate. Therefore, I would like to ask the authors whether it would be possible to use some of the knowledge from the life of wild bird species, e.g. also in tropical areas, and possibly apply it in the future for use in domestic poultry species. I did not find similar considerations in this article.
Author Response
Reviewer(s)' Comments to Author:
Reviewer: 2
1) This article is based on the current problems caused by increasingly long and intense periods of high air temperatures in many areas of the world. From this perspective, poultry breeding and fattening are becoming increasingly demanding in terms of deeper understanding of the related issues and the use of new knowledge in practice.
Answer 1(A1): Thank you for your comments.
2) The article is written in a clear manner and brings a number of new findings that may be very useful for readers of the Animals journal who deal with the given issue or are focused on the related sectors of poultry breeding and animal production.
A2: Thank you for your comments.
3) I particularly appreciate part 5. Strategies for mitigating heat stress splenic injury. I would recommend that the authors try to supplement and further develop this important part of their article in the future.
A3: Thank you for your comments. In the fifth part of the revised draft, we continue to expand the relevant content of the collaborative application of heat adaptation training and genetic improvement in the prevention of heat stress-induced spleen injury in poultry. See line 399-422.
4) Similarly, in my opinion, the last part of the article 6. Outlook is also very valuable, in which the authors indicate the direction in which further development and research in this area should take place.
A4: Thank you for your comments. We added some content at the end of the sixth part, hoping to make it more complete. See line 467-469.
5) Overall, I evaluate this article very positively as beneficial for the given scientific field and related fields. Heat stress in a changing environment also affects wild birds, which can survive even in a very hot climate. Therefore, I would like to ask the authors whether it would be possible to use some of the knowledge from the life of wild bird species, e.g. also in tropical areas, and possibly apply it in the future for use in domestic poultry species. I did not find similar considerations in this article.
A5: Thank you very much. Your suggestion is a great inspiration to us. In order to prevent the heat stress of birds in the future, heat acclimation can also be used to improve the heat resistance of poultry, and then select heat resistant varieties. The use of combinatorial genomics to screen different genes in the spleen of tropical birds and ordinary poultry is helpful to cultivate poultry populations that adapt to the climate, while minimizing the loss of productivity. See line 410-422.